# Parental Self-Efficacy and Child Diet Quality between Ages 2 and 5: The STEPS Study

**DOI:** 10.3390/nu14224891

**Published:** 2022-11-18

**Authors:** Saija Tarro, Mirkka Lahdenperä, Niina Junttila, Antti Lampimäki, Hanna Lagström

**Affiliations:** 1Department of Public Health, University of Turku and Turku University Hospital, FI-20014 Turku, Finland; 2Centre for Population Health Research, University of Turku and Turku University Hospital, FI-20014 Turku, Finland; 3Department of Biology, University of Turku, FI-20014 Turku, Finland; 4Department of Teacher Education, University of Turku, FI-20014 Turku, Finland; 5Department of Teacher Education, University of Jyväskylä, FI-40014 Jyväskylä, Finland

**Keywords:** diet quality, parental self-efficacy, pre-school children

## Abstract

Parental self-efficacy (PSE), a measure of the subjective competence in the parental role, has been linked with child well-being and health. Research on the influence of PSE on child eating habits is scarce, and the few studies have concentrated on certain food groups, such as vegetables or fruits, and have mostly included only maternal PSE. Thus, the aim of this study was to explore the associations between PSE (separately for mothers and fathers and as a total family-level score) and child diet quality in a cross-sectional and longitudinal study setting. PSE was measured at child ages of 1.5 and 5 years, and diet quality was measured at ages 2 and 5. Participants are from the Steps to Healthy Development (STEPS) Study (*n* = 270−883). We found that maternal PSE and family level PSE score were associated with child diet quality. Paternal PSE was not, but the dimension Routines was associated with child diet quality. PSE was similarly associated with child diet quality at both age points. Our results suggest that PSE is an important construct in the development of healthy dietary habits in children, and supporting parenting programs aimed at higher PSE could promote healthy diet quality in children.

## 1. Introduction

Nutrition in early life, especially during the first 1000 days, influences the health of the adult and the risk of non-communicable diseases later in life [1,2,3]. Dietary habits establish largely during the first 2 years of life, starting from the innate preference for sweet tastes to the acceptance of new foods through the repeated exposure to different tastes [4]. Dietary habits for young children are mainly influenced by parents as they provide food and shape food environments by choosing which foods are available at home and served at the dinner table. [5,6,7].

Parents have a generally good knowledge of the dietary recommendations [8,9,10]. However, the knowledge of good nutrition can more likely be transformed to practice if parents are confident in their ability to positively influence the dietary habits of their children [11,12]. A central concept under social cognitive theory, self-efficacy, refers to the subjective feelings of one’s ability to successfully perform a particular behavior [13]. Parental self-efficacy (PSE) refers to a parent’s appraisal of his or her competence in the parental role and perceptions of their ability to positively influence the development and behaviors of their children [12,14].

PSE has been linked with child lifestyle factors, such as physical activity, dietary intakes of certain food groups, such as vegetables or fruits [15,16,17,18,19,20,21], and child food preferences [22]. A higher PSE has been associated with higher fruit and vegetables intake [16,17,18,23,24] and a lower intake of non-core foods [19], such as sweets and sodas [18,24] and unhealthy snacks [15]. In addition, one study has shown an association between maternal self-efficacy and child diet quality [25].

However, the previous results are based mostly on cross-sectional findings on maternal self-efficacy [19,20,22,23,24,25]. To date, only one study investigated the role of paternal self-efficacy in child food consumption, suggesting that there is a positive association between paternal self-efficacy and children’s fruit and vegetables consumption [26]. Further, to our understanding, no previous studies have included a combination score for both parents’ self-efficacy to examine the effects of both caregivers together on the child’s dietary habits.

In the present study, we investigated the association between PSE (at 1.5 and 5 years of age) and the diet quality of children at the age of 2 and 5 years. We also investigated if these associations change with the child’s age with a longitudinal study setting. The associations of PSE with the diet quality of children were examined separately for mothers (mother’s PSE) and fathers (father’s PSE) and as a combination score for both parents (family-level PSE).

## 2. Materials and Methods

### 2.1. Study Design and Subjects

The present study is based on data from a longitudinal Finnish follow-up study, Steps to Healthy Development of Children (the STEPS Study), which has previously been described in detail elsewhere [27]. Briefly, all Finnish- and Swedish-speaking mothers who delivered a living child between 1 January 2008 and 31 March 2010 in the Hospital District of Southwest Finland formed the cohort population (*N* = 9811 mothers and their 9936 children). A total of 1797 mothers, (18.3% of the total cohort) and 1658 fathers with 1805 neonates (30 pairs of twins) volunteered as participants for the follow-up group of the STEPS study.

Children born at full-term (after 36 ^6/7^ weeks of pregnancy) from singleton pregnancies were included in the present study (*n* = 1683). Parents who were not divorced by age 5 of the child, were included (*n* = 1609). The number of participants varied between the study questions based on questionnaire data availability. Child diet quality and PSE were available for 575 families at a 2-year age point, for 396 families at a 5-year age point and for 270 families at both age points (Table 1). There were no differences between dropout families between 2- and 5-year age points and non-dropouts (Appendix A Appendix A).

The STEPS study has been approved by the Ethics Committee of Hospital District of Southwest Finland (2/2007). Written informed consent was obtained from all the participants in the study and/or their parents.

### 2.2. Outcome Variable: Child Diet Quality

The parents (mother or father) reported their children’s dietary habits with a short questionnaire when the child was 2 and 5 years old. The questionnaire was described in detail in our previous article [28]. Briefly, 10 questions concerning dietary habits were selected according to the Finnish dietary recommendations for children at the time of the data collection [29]. The questions were as follows: (1) how many times did the child eat breakfast per week, (2) how many times per day did the child eat after breakfast (meals and snacks included), (3) the type of drink with meals, (4) the quality of milk, (5) primary beverage, (6) the type of fat used on bread, (7) how many times did the child eat fish per week, (8) how many portions of vegetables did the child eat per day, (9) how many portions of fresh fruits and berries did the child eat per day and (10) how many times did the child eat unhealthy salty or sweet snacks per week. Each recommended choice provided one point for the diet quality score, so the overall score varied from 0 to 10, the higher values indicating a greater adherence to the recommendations [28]. The diet quality score was used as a continuous variable in the analyses.

### 2.3. Parental Self-Efficacy

The parents (both mother and father) evaluated their PSE using the domain-specific scale of parental self-efficacy when their child was at age 1.5 and 5 years. Only parents who were not divorced were included (74 parents were excluded); see Appendix A Appendix A. The scale is a modified Finnish version of the Self-Efficacy for Parenting Tasks Index Toddler Scale (SEPTI-TS) [30]. The scale was validated within the STEPS study in Junttila et al. 2015 [31], including details about the PSE questionnaire. The validated scale includes five dimensions of PSE: Presence, Emotional support, Routines, Playing and Teaching. Presence refers to the ability to be present for a child (e.g., “When my child needs me, I am able to interrupt what I am doing.”), Emotional Support describes how sensible the parent is to the child’s emotional needs (e.g., “My child knows that I understand when her/his feelings have been hurt.”), Routines refer to the parents’ estimation of their ability to reasonably arrange their children’s daily life (e.g., “I have managed to create daily routines that are suitable for me, as well as for my child.”), Playing describes the amount and the quality of joined play activities between the parent and child (e.g., “I easily invent different games with my child.”) and Teaching describes the quality of joined teaching and learning experiences between the parent and child (e.g., “I believe that my child learns a lot while I show her/him examples and teach her/him things.”). The total number of questions was 20, and each dimension consisted of four items and was answered on a Likert scale from 1 to 5. The maximum score, estimating the best possible PSE for each dimension, is 20. To create a composite score for mother’s and father’s PSE on separate dimensions, the items were summed up when all the dimension scores were available (max value 20 per dimension, 100 for total maternal or paternal scale). Further, a composite score for family-level PSE was created by taking a mean of the mother’s and father’s PSE total scores (max value 100).

### 2.4. Covariates

Based on earlier literature [32,33,34,35], the following variables affecting children’s dietary quality and parental self-efficacy were selected as covariates (1) age of parents, (2) family income, (3) mother’s and father’s education and (4) number of siblings. Information regarding the mother’s and father’s age and education was obtained from self-administered questionnaires upon recruitment during pregnancy. The mother’s age was classified into two categories by the mean age of women giving birth in Finland 2019 (29.6 years of age) [36]. The same cut-off age was used with the father’s age. The mother’s and father’s ages were used as continuous variables in the analyses and as categorical variables in Table 2. Information regarding the total family income (after taxes) and the number of siblings was obtained from self-administered questionnaires at the child’s age of 2 and 5 years. Total family income (including both parents) was measured with a five-point scale (under EUR 1000, EUR 1000–2000, EUR 2000–3000, EUR 3000–4000 and over EUR 4000 per month). The family income was then divided into two categories: under EUR 3000 and EUR 3000 or higher per month.

Parental education variables (separately for mothers and fathers) were classified as advanced education or low education. Those who had no professional training or a maximum of an intermediate level of vocational training were classified as “low” (no education, vocational courses/apprenticeship training, vocational upper-secondary education or vocational college). Those who had studied at a university or a University of Applied Sciences were classified as “advanced” (bachelor’s degree, master’s degree, licentiate or doctoral degree). In the analyses with family-level PSE, a family education variable was used based on the highest education that one of the parents had completed for their professions.

Family education and family income were used separately to indicate family socioeconomic status (SES).

### 2.5. Statistical Analysis

The associations between child diet quality and sociodemographic variables were explored using independent t-tests separately at 2 and 5 years of age. Paired t-tests were used to test the similarity of diet quality between 2 and 5 years. Similarly, the difference in PSE variables between ages 1.5 and 5 years were tested with paired t-tests.

Linear regression models were used to model the associations with PSE at 1.5 years and child diet quality at 2 years of age and PSE at 5 years and child diet quality at 5 years of age. PSE variables (mother’s PSE, father’s PSE and family-level PSE) were used as continuous explanatory variables in the models, and child diet quality was the continuous outcome variable. Separate models were run for the mother’s PSE, the father’s PSE, the family-level PSE and for each PSE dimension (Presence, Emotional Support, Routines, Playing, Teaching). All models were adjusted for sociodemographic factors (child sex, mother’s age, father’s age, number of siblings, mother and father education or family education and family income).

Generalized linear mixed models were used to model the longitudinal associations of mother’s PSE, father’s PSE, family-level PSE, all five PSE dimensions and diet quality. The models included the child’s age (1.5/2 and 5 years) and an age–self-efficacy interaction term to investigate if the associations with PSE and child diet quality change with the child’s age (time). Child id was used as a repeated term to control for the intraindividual correlation between repeated measurements. The models were adjusted for sociodemographic factors (sex, mother’s age, father’s age, number of siblings, mother and father education or family education and family income). For longitudinal analysis, sample dropout analysis was conducted to check the differences between the 2-year age point sample and the longitudinal sample (including both 2- and 5-year age points).

Statistical analysis was performed using SAS software for Windows version 9.4 (SAS Institute Inc., Cary, NC, USA). The level of significance was set at a *p*-value < 0.05.

## 3. Results

### 3.1. Description of Participants

The sociodemographic characteristics of the families in relation to diet quality are presented in Table 2. The child diet quality score was higher, 6.48 (standard deviation (SD) = 1.63), at 5 years, compared to 6.10 (SD = 1.69) at 2 years (*p* < 0.001). Overall, those children with a better diet quality score at the age of 2 and 5 years were characterized by a family with a high income (*p* = 0.007) and high education (*p* < 0.001) level. Both mothers’ and fathers’ education were positively associated with the child’s diet quality score at both age points. In addition, fewer siblings at the age of 2 years were associated with a higher child diet quality score at the same age (Table 2).

### 3.2. Parental Self-Efficacy

Descriptive characteristics of the PSE at both age points are presented in Table 3. The family-level PSE was higher (84.4 (SD = 6.7)) at 1.5 years compared to what it was (81.9 (SD = 6.8)) at 5 years (*p* < 0.001). Similarly, mother’s PSE and father’s PSE were higher at a child age of 1.5 years compared to 5 years (*p* < 0.001). Further, the family-level PSE was higher at 1.5 years in relation to the Presence, Playing and Teaching dimensions (*p* < 0.001). The mother’s and father’s Presence dimension was higher at 1.5 years compared to 5 years of age (*p* < 0.001 for both the mother and father). Similarly, the Playing (*p* < 0.001 for both the mother and father) and Teaching (*p* = 0.004 for mothers and *p* < 0.001 for fathers) dimensions were higher at 1.5 years. The dimension of Emotional support was the only stable dimension for both mothers (*p* = 0.24) and fathers (*p* = 0.11) and for family-level PSE (*p* = 0.73). The dimension Routines was stable for mother’s and father’s PSE, but when looking at the family-level PSE, there seemed to be a slight increase in that dimension from 1.5 years to 5 years (*p* = 0.03).

### 3.3. Association of Diet Quality with Family-Level PSE

The cross-sectional associations of family-level PSE at 1.5 and 5 years of age with diet quality at 2 and 5 years of age are shown in Figure 1 and Table 4. Family-level PSE was positively associated with child diet quality at both age points, meaning that the higher the family-level PSE, the higher the child diet quality (Figure 1 and Table 4). Family-level Routines were positively associated with child diet quality at both age points, along with Teaching at the 2-year age point and Emotional support and Playing at the 5-year age point.

The longitudinal analyses with PSE confirmed the results from the cross-sectional analyses (Table 5). Family-level PSE was positively associated with child diet quality. In addition, there was a positive association between the Presence, Routines, Playing and Teaching dimensions and child diet quality. Additionally, age was positively associated with diet quality, as it was in the cross-sectional analyses, meaning that diet quality was better at 5 years than it was at 2 years. However, family-level PSE and the dimensions were as similarly associated with child diet quality at both age points (interactions were nonsignificant).

### 3.4. Association of Diet Quality with Mother’s PSE

Cross-sectional associations of mother’s PSE at 1.5 and 5 years of age with diet quality at 2 and 5 years of age are shown in Table 4. A clear association was seen between mother’s PSE and child diet quality in all dimensions, except for the Emotional support at 2 years of age and Presence at 5 years of age. The higher the score for the mother’s PSE or dimensions, the higher the child diet quality score (Table 4).

The longitudinal analyses of mother’s PSE with child diet quality showed somewhat similar results with the cross-sectional analyses (Table 5). There was a positive association between mother’s PSE and child diet quality in all dimensions. In addition, age–PSE interaction was statistically significant with the mother’s Presence dimension, meaning that Presence was more strongly associated with child diet quality at 2 years than at 5 years: estimate 0.10 (standard error (SE) 0.03) at 2 years vs. estimate 0.03 (SE 0.03) at 5 years.

### 3.5. Association of Diet Quality with Father’s PSE

Cross-sectional associations of father’s PSE at 1.5 and 5 years of age with diet quality at 2 and 5 years of age are shown in Table 4. Father’s PSE was not associated with child diet quality, but only Routines at both age points and Teaching at 2 years of age were. The higher the score for father’s Routines and Teaching related to PSE, the higher the child diet quality score (Table 4).

The longitudinal analyses of father’s PSE with child diet quality similarly showed an association between the father’s Routines dimension and child diet quality (Table 5). However, the father’s Routines dimension was similarly associated with child diet quality at both age points (interactions were nonsignificant).

## 4. Discussion

Our main findings were that family-level PSE and mother’s PSE were positively associated with the child diet quality. For mothers, the association was visible with the Routines, Playing and Teaching dimensions at both age points. For fathers, the association was seen only with the Routines dimension at both age points. The longitudinal analyses confirmed these findings and suggested that the family-level PSE is an important modifiable factor affecting child dietary habits.

In this study, we saw the declining trend of PSE in all dimensions except for Emotional support and Routines. This is consistent with an earlier study [31] which found that parents evaluated their PSE stronger while their child was 1.5 years old compared with 3 years old. Our finding was similar to a previous study concluding that mothers of 5-year-old children were less confident than mothers of 1-year-old children regarding their ability to impact child healthy eating [19]. In addition, this finding is consistent with the view that self-efficacy cannot be seen as a fixed personality trait but merely as a construct which is dynamically influenced by the parent’s individual variables such as emotional status, the child’s variables such as temperament and family factors such as perceived marital support [31,37].

Our results are in line with previous studies suggesting that PSE is associated with healthier dietary habits in children [15,16,17,18,19,20,24]. Most of the previous research has been conducted with mothers. According to our knowledge, this is among the first studies to measure mother’s PSE and father’s PSE separately and to include a combination score from the mother and father. Our results show a clear association between family-level PSE and child diet quality. The association was also seen in the Routines and Teaching dimensions at 2 years of age and in the Emotional support, Routines and Playing dimensions at 5 years of age. When looking at the data in a longitudinal setting, similar associations were found, confirming the results. Regardless of the smaller sample size in the longitudinal analysis, family-level PSE and all dimensions, except for Emotional support, were strongly associated with child diet quality. Based on the results, the associations with PSE and child diet quality are clearest with mothers. Thus, the family-level PSE and child diet quality associations might merely reflect the maternal effects on the associations.

What surprised us was that the association was so clearly visible only with mother’s PSE and child diet quality, and father’s PSE did not seem to associate with child diet quality on a general level. However, the dimension Routines was clearly associated with child diet quality with fathers as well. The findings are important, as a previous study suggests that a father’s parenting behaviors and styles may be associated with an increased risk of preschooler overweightness [38]. Based on the earlier research, fathers seem to be less responsible for child feeding [39] than mothers, even in modern Western societies, and mothers are more involved with childcare and parenting activities than fathers [40,41]. According to the Finnish Institute of Health and Welfare, the division of household tasks still remains traditional in Finland, and Finnish mothers are still more likely to be responsible for cooking than men [41]. Thus, the associations with father’s self-efficacy and child diet quality should be further studied.

A more detailed analysis of PSE and child diet quality was possible through a separate analysis of PSE dimensions and child diet quality. According to the results, PSE was similarly associated with child diet quality at both age points. The only exception was mothers’ Presence dimension, which was more strongly associated with child diet quality at 2 years of age than at 5 years of age. Most of the children (89%) from the study population attended daycare at 5 years of age [42]. As our results also show, the diet quality of 5-year-old children was better than that of 2-year-old children. The ability of working mothers to allocate time towards children’s nutritional needs is limited [43]. However, daycare meals in Finland are of high quality and should follow national dietary recommendations [29], and that is visible in the total diet quality of 5-year-old children. Thus, other factors might play a more important role at this age point.

In our study Mother’s Emotional support was associated with child diet quality, especially at 5 years of age. This dimension might reflect emotional eating in children. There is some indication that children of parents with a low PSE might be prone to emotional eating styles [44]. Parents might also use emotion regulation feeding when responding to their child’s negative feelings by giving them food [45]. A greater use of food for comfort may predict greater child emotional eating [46]. We found earlier that child emotional undereating at the age of 5 years is associated with lower child diet quality [28]. Our findings are supported by a recent study showing that emotional undereating might be negatively associated with adherence to a Mediterranean diet in school-age children [47].

Routines was the dimension which was most clearly associated with child diet quality for mother and father and family-level PSE. The Routines dimension might reflect partly regular meal patterns in the family. According to Finnish nutrition recommendations for young children, one important part of good nutrition is regular meals [29]. Previous studies have indicated that meal skipping might be associated with lower diet quality with children under school age [48,49]. A recent population-based prospective cohort study suggests that family mealtime routines in early childhood may promote child whole diet quality later in childhood [49]. Parents who eat together with the children are able to model eating behaviors and maintain family food rules [50,51]. Household food rules, such as “no meals while watching television”, have been found to be effective in maintaining better diet quality among adolescents [52].

Parental self-efficacy can also be seen as a parent’s confidence to make good decisions for their children and confidence in the parent’s ability to exert control over the child’s behavior [53,54]. Therefore, PSE has also been linked with more general parenting styles [54,55], and some results suggest that an authoritative parenting style would be associated with a higher PSE [54]. Further, authoritative parenting has been linked with a favorable child diet [56,57]. Previous studies have also found an association between PSE and food parenting practices, such as controlling feeding practices [58,59]. Mothers with higher self-efficacy might be using health-promoting child feeding practices [59]. Thus, PSE is an important parental level concept affecting child dietary behavior.

It is especially interesting that we were able to show associations between PSE and child diet quality using a broader scale on PSE than previous studies. Except for one study [24], most of the previous studies have used these “task-specific” self-efficacy measures [15,16,17,18,19,20]. According to our knowledge, this is the first time the “domain-specific” PSE scale [60] is used to study the association between PSE and child dietary habits. Parental self-efficacy can be assessed and measured at different levels of specificity: global, domain- and task-specific self-efficacy [30,37,55]. Task specific efficacy beliefs refer to some very specific tasks, such as getting the child to eat enough vegetables or fruits. Domain-specific efficacy measures combine different task-specific measures of self-efficacy, resulting in a multidimensional index of domain-level parenting self-efficacy [30]. In global measures, the assessment is based on global competence expectations which are not linked to particular tasks but might act as predictors of general parental qualities, such as sensitivity, warmth, etc. [12]. Several “task-specific” scales have been developed to measure PSE specifically in relation to children’s dietary habits [19,61,62,63]. The domain-specific SEPTI-TS scale we have used was originally developed by Coleman & Karraker (2003), takes into consideration the developmental phase of a child and was specifically designed to measure the parenting of toddlers, which might be a rather difficult phase of parenting.

The large sample size and amount of data from both parents are major strengths of the study. In addition, we have assessed several sociodemographic and family-related factors affecting child dietary habits and included several covariates in the analysis. This is the first study to investigate the associations between parental self-efficacy from both parents and the whole diet quality of a child. Lastly, the longitudinal study setting gives a more comprehensive overview on the associations during early childhood.

This study has also some limitations. The child diet quality was assessed by individual questions that were partly derived from the adult validated questionnaire “Index of Diet Quality” [64]. However, it can be assumed that the questions derived from the Finnish nutritional recommendations for infants and young children [29] proved at least a reasonable overview of the child diet quality. In addition, we cannot put aside self-report bias, as it may be possible that parents provided socially desirable responses. In addition, PSE was measured at the child age of 1.5 years, and diet quality was measured at 2 years of age. However, according to previous studies, PSE is fairly stable within such a short time period [31,65].

## 5. Conclusions

Our results suggest that PSE is an important construct in the development of healthy dietary habits in children. These results indicate that PSE would act as an important target for family interventions supporting a child’s healthy eating. Additionally, supporting parents during early childhood, when their self-efficacy is higher, might increase the parents’ capacity to provide a health-promoting diet for their children. Increasing collaboration between nutritionists and behavioral scientists with personal working in well-baby clinics or day-care centers might offer new tools to enhance child dietary habits. In addition, the results imply that PSE is an important measure to be considered in future studies, especially including father’s PSE. However, further studies are needed in other study populations and countries, as the beneficial effect of PSE on childhood diet may depend on different parenting styles and differences in food cultures.

## Figures and Tables

**Figure 1 nutrients-14-04891-f001:**
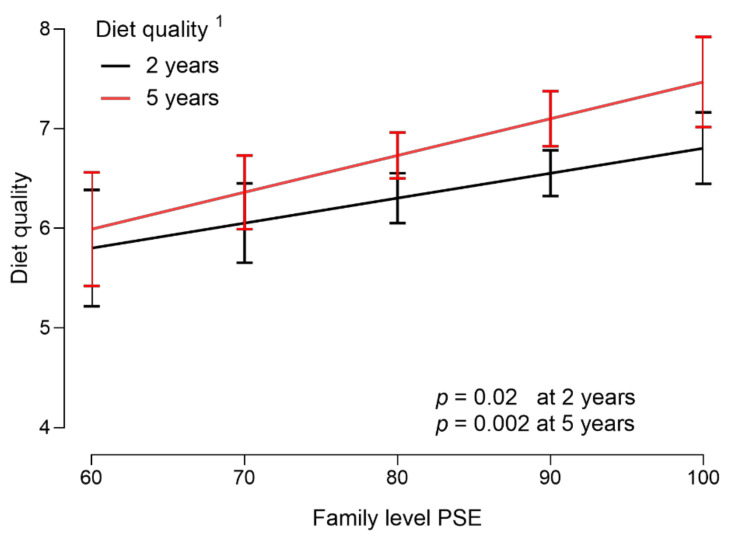
Child diet quality at 2 and 5 years of age and family-level PSE at 1.5 and 5 years of age (*n* = 538 at 2 years and *n* = 392 at 5 years). ^1^ Mean values with 95% confidence intervals adjusted for covariates (mother’s age = 30.8 and father’s age = 32.9, family income ≥ EUR 3000, family education = Advanced education, number of siblings 0–1). Statistical method: Linear regression models. PSE, parental self-efficacy.

**Table 1 nutrients-14-04891-t001:** Study population and sample sizes at each age point.

	Study Sample	2 Years	5 Years	2 and 5 Years
	*N*	*N*	*N*	*N*
Child diet quality	1683 *	785	645	499
Family-level PSE		762	462	371
Mother’s PSE	1797	883	725	593
Father’s PSE	1658	770	544	429
Family-level PSE + Diet quality		575	396	270
Mother’s PSE + Diet quality		665	597	411
Father’s PSE + Diet quality		581	416	286

* Original sample size (*n* = 1805); preterm infants and twins excluded. PSE, parental self-efficacy.

**Table 2 nutrients-14-04891-t002:** Sociodemographic characteristics of the study participants and diet quality with means and standard deviations (SD) ^1^.

Variable		Diet Quality ^2^					
		2 Years			5 Years		
		*n* (%)	Mean (SD)	*p*	*n* (%)	Mean (SD)	*p*
All		575	6.10 (1.69)		396	6.48 (1.63)	<0.001
Sex	Boy	308 (54%)	6.17 (1.76)	0.34	206 (52%)	6.52 (1.63)	0.64
	Girl	267 (46%)	6.03 (1.60)		190 (48%)	6.45 (1.64)	
Mother age							
	17–29	227 (40%)	6.15 (1.61)	0.55	140 (35%)	6.66 (1.58)	0.13
	30–45	347 (60%)	6.06 (1.74)		256 (65%)	6.39 (1.65)	
Father age							
	17–29	155 (27%)	5.97 (1.68)	0.25	103 (26%)	6.50 (1.64)	0.96
	30–45	420 (73%)	6.15 (1.69)		293 (74%)	6.48 (1.63)	
Family income							
	EUR <3000	297 (54%)	5.91 (1.69)	0.005	135 (34%)	6.18 (1.62)	0.007
	EUR ≥3000	248 (46%)	6.32 (1.67)		258 (66%)	6.65 (1.62)	
Mother education							
	Low	181 (32%)	5.77 (1.69)	<0.001	117 (30%)	6.01 (1.59)	<0.001
	Advanced	381 (68%)	6.29 (1.65)		276 (70%)	6.68 (1.62)	
Father education							
	Low	267 (48%)	5.90 (1.62)	0.007	180 (46%)	6.17 (1.57)	<0.001
	Advanced	293 (52%)	6.28 (1.71)		210 (54%)	6.77 (1.64)	
Family education ^3^							
	Low	132 (23%)	5.60 (1.66)	<0.001	83 (21%)	5.75 (1.47)	<0.001
	Advanced	436 (77%)	6.26 (1.67)		312 (79%)	6.68 (1.62)	
Number of siblings							
	0–1	479 (83%)	6.20 (1.68)	0.002	274 (69%)	6.57 (1.59)	0.12
	2 or more	96 (17%)	5.63 (1.68)		122 (31%)	6.30 (1.71)	

^1^ Statistical difference were tested with t-tests. The data include only families who have all questionnaire data available (maternal and paternal SE and child diet quality at the specific time point). ^2^ Mean score for adherence to the Finnish dietary recommendations for children in 2004; total points based on 10 individual dietary items for the dietary score. The range of the diet quality score varied between 1 and 10 points in children. ^3^ Highest education that one of the parents had completed for their professions. Those who had no professional training or a maximum of an intermediate level of vocational training were classified as “low”. Those who had studied at a University of Applied Sciences or higher were classified as “advanced”. SD, standard deviation; SE, self-efficacy.

**Table 3 nutrients-14-04891-t003:** Comparisons of the mean of the family-level PSE, mother’s and father’s PSE and their dimensions at 1.5 and 5 years of age.

Variable		1.5 Years	5 Years	
	*n*	Mean (SD)	Mean (SD)	*p* ^1^
Family-level PSE	371	84.4 (6.7)	81.9 (6.8)	<0.001
Presence		17.3 (1.6)	16.4 (1.6)	<0.001
Emotional support		17.4 (1.4)	17.4 (1.5)	0.73
Routines		16.9 (1.9)	17.1 (1.9)	0.032
Playing		15.6 (2.2)	14.3 (2.4)	<0.001
Teaching		17.2 (1.6)	16.8 (1.6)	<0.001
Mother’s PSE	593	85.7 (8.0)	83.1 (9.0)	<0.001
Presence		17.5 (2.1)	16.5 (2.5)	<0.001
Emotional support		17.8 (1.9)	17.9 (1.9)	0.24
Routines		17.4 (2.2)	17.6 (2.3)	0.06
Playing		15.4 (3.0)	13.8 (3.4)	<0.001
Teaching		17.4 (2.1)	17.2 (2.2)	0.004
Father’s PSE	429	83.1 (9.2)	80.9 (8.9)	<0.001
Presence		17.1 (2.3)	16.2 (2.4)	<0.001
Emotional support		17.0 (2.1)	16.8 (2.0)	0.11
Routines		16.2 (2.6)	16.5 (2.4)	0.09
Playing		15.8 (2.9)	15.0 (3.0)	<0.001
Teaching		17.0 (2.3)	16.4 (2.3)	<0.001

^1^ Statistical differences between time points were tested with paired t-tests. The data include only families who have PSE questionnaire data available from both age points.

**Table 4 nutrients-14-04891-t004:** Associations between PSE at 1.5 years and 5 years of age and child diet quality at 2 and 5 years of age.

Variable	Diet Quality ^1^					
	2 Years			5 Years		
	*n*	Estimate (95 % CL) ^2^	*p*	*n*	Estimate (95 % CL) ^2^	*p*
Family-level PSE	538	0.24 (0.04–0.45) ^3^	0.02	392	0.37 (0.14–0.60) ^3^	0.002
Presence		0.08 (−0.03–0.16)	0.06		0.07 (−0.02–0.15)	0.14
Emotional support		−0.02 (−0.12–0.07)	0.64		0.14 (0.04–0.24)	0.008
Routines		0.09 (0.02–0.17)	0.01		0.16 (0.08–0.24)	<0.001
Playing		0.06 (−0.006–0.12)	0.08		0.08 (0.02–0.15)	0.01
Teaching		0.12 (0.04–0.21)	0.004		0.08 (−0.02–0.18)	0.098
Mother’s PSE	615	0.32 (0.16–0.49) ^3^	0.001	580	0.24 (0.09–0.39) ^3^	0.002
Presence		0.10 (0.04–0.16)	0.002		0.04 (−0.01–0.10)	0.13
Emotional support		0.04 (−0.03–0.11)	0.31		0.07 (0.009–0.14)	0.03
Routines		0.08 (0.02–0.14)	0.009		0.11 (0.05–0.16)	<0.001
Playing		0.05 (0.009–0.10)	0.02		0.04 (0.006–0.08)	0.02
Teaching		0.11 (0.05–0.18)	<0.001		0.07 (0.01–0.13)	0.02
Father’s PSE	537	0.09 (−0.07–0.24) ^3^	0.28	389	0.14 (−0.04–0.32) ^3^	0.13
Presence		0.01 (−0.05–0.07)	0.85		0.02 (−0.04–0.09)	0.47
Emotional support		−0.03 (−0.10–0.04)	0.42		0.03 (−0.05–0.11)	0.44
Routines		0.06 (0.01–0.11)	0.03		0.09 (0.02–0.15)	0.01
Playing		0.03 (−0.02–0.07)	0.30		0.03 (−0.02–0.08)	0.26
Teaching		0.07 (0.002–0.13)	0.04		0.02 (−0.05–0.09)	0.62

^1^ Mean score for adherence to the Finnish dietary recommendations for children in 2004; total points based on 10 individual dietary items for the dietary score. The range of diet quality score varied between 1 and 10 points in children. ^2^ Adjusted for the sex of the child, the mother’s age, the father’s age, family income, mother’s, father’s or family education and the number of siblings. Statistical method: Linear regression models. ^3^ Estimate calculated per 10 PSE points. CL, Confidence Level.

**Table 5 nutrients-14-04891-t005:** Associations between PSE at 1.5 years and 5 years of age and child diet quality at 2 and 5 years of age in a longitudinal study setting.

	Estimate ^1^	*p*-Value		
Model	PSE	PSE	Age	Age × PSE Interaction
Family-level PSE (*n* = 270)	0.340 ^2^	<0.001	<0.001	0.57
Presence	0.065	0.05	<0.001	0.79
Emotional support	0.069	0.07	<0.001	0.27
Routines	0.14	<0.001	<0.001	0.52
Playing	0.069	0.006	<0.001	0.38
Teaching	0.098	0.006	<0.001	0.63
Mother’s PSE (*n* = 411)	0.265 ^2^	<0.001	<0.001	0.14
Presence	0.054	0.02	<0.001	0.04
Emotional support	0.066	0.009	<0.001	0.63
Routines	0.089	<0.001	<0.001	0.84
Playing	0.043	0.007	<0.001	0.43
Teaching	0.059	0.01	<0.001	0.28
Father’s PSE (*n* = 286)	0.104 ^2^	0.11	<0.001	0.52
Presence	0.014	0.58	<0.001	0.70
Emotional support	−0.004	0.88	<0.001	0.34
Routines	0.081	<0.001	<0.001	0.76
Playing	0.009	0.67	<0.001	0.32
Teaching	0.034	0.19	<0.001	0.81

^1^ Adjusted for the sex of the child, the mother’s age, the father’s age, family income, family education and the number of siblings. Statistical method: Generalized linear mixed models. ^2^ Estimate calculated per 10 PSE points.

## Data Availability

All data for the STEPS Study are collected, managed, maintained, and owned by the University of Turku. Collaboration is encouraged, but the data are still being collected and are not currently available for access by other researchers. The cohort dataset cannot be made openly available due to confidentiality in order to protect the cohort participants’ identity. Due to the longitudinal nature of the STEPS study, the data cannot be fully anonymized in the near future. Data sharing outside the study group requires a data-sharing agreement. Investigators can submit an expression of interest to the STEPS study’s Steering Committee. Researchers interested in collaborative work or further information are invited to contact the principal investigator, Hanna Lagström (email: hanlag@utu.fi).

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
