# Peer review of "Parental Self-Efficacy and Child Diet Quality between Ages 2 and 5: The STEPS Study"

_nutrients, 2022, doi:10.3390/nu14224891_

Round 1

Reviewer 1 Report

This is my review on Parental Self-Efficacy and Child Diet Quality between 2 and 5 years. The STEPS Study.

Introduction is fine. The aim of the study is clearly presented. Materials and results are thoroughly described and presented in comprehensive way. 

I agree with strengths and limitations presented. Discussion is interesting.

The findings are worth publishing. This is a fine paper. 

Minor English revisions are required.

Author Response

Response to comments from the Reviewer 1

We thank the Reviewer for a very positive attitude towards our paper.

We have now done a language check for the article with a native speaker.

Reviewer 2 Report

Dear Authors,

I would like to congratulate you on the great work you have presented. Infant feeding has been and will always be a very important aspect of research in maternal and child nutrition. Self-efficacy is something that has been less explored; be it in feeding, food quality, or breastfeeding. Therefore, yourr work is quite interesting to those who are in the area.  I also appreciate the concept of using the existing data from STEPS which shows that sometimes researchers just need to look around to find out what to do with existing data.

As such, I do not have major comments on your manuscript; however, here are a few areas that I believe would greatly improve the presentation of your work.

Methods section:

Please attach the questionnaire (in the appendix) in the methods section so other readers can relate to your study and can possibly use or adapt the tool in another setting.

Statistical analysis:

1.       You have used the following words “the cross-section association of the child diet quality…” instead use the standard epidemiological terms. The association between X and Y was explored using an independent t-test.

Results:

Your result is quite dense and sometimes it is hard to follow. I would suggest breaking it down into small sub-sections. Examples could be:

·       Description of participants

·       Diet quality and self-efficacy

·       Association of diet quality…(use three distinct paragraphs to highlight: maternal; paternal and family level).

This also means re-arranging your result section but it will make your paper read much better.

Presentation of Table:

Great work; and very informative tables. I like the way you have presented the findings. Easy to read and indicate the significant results. My only concerns are about your table heading. Table 1; is great with a short title. Table 2- 5 has a long table title. It feels like we are reading a footnote instead of a table. Please use a short table title. Additional information such as ‘adjust for’..’ can be put as the footnote. This will make your hard work look better.  

Figure 1: Please use a similar approach for Figure 1 too.

Discussion:

The discussion is dense and useful for the readers in the field. Again, consider restructuring the discussion once you re-align results.

I think you have done great work on this paper. Consider adding a few lines on how your findings can be used by those who are at the implementation and policy levels.

Overall, the great manuscript I read in a while. I wish all the best to the authors.

Author Response

Response to comments from the Reviewer 2

We thank the Reviewer for a very positive attitude towards our paper. We also thank you for the valuable comments for the improvement of our manuscript. Please find below first the comments as they were raised by the Reviewer, then our response, and if necessary, the text as it reads in the revised manuscript.

We wish all the best to You too!

Comment: Please attach the questionnaire (in the appendix) in the methods section so other readers can relate to your study and can possibly use or adapt the tool in another setting.

Our response:

Thank you for the suggestion. However, as the PSE questionnaire has been previously published we cannot attach it to this article, but it can be viewed from the article where it was originally published (Junttila et al. 2015, https://doi.org/10.1111/fare.12161).

Comment: You have used the following words “the cross-section association of the child diet quality…” instead use the standard epidemiological terms. The association between X and Y was explored using an independent t-test.

Our response:

Thank you for the comment. We have now revised the sentence in the statistical analysis section as follows:

“The association between child diet quality and sociodemographic variables were explored using independent t-tests separately at 2 and 5 years of age.”

Comment: Your result is quite dense and sometimes it is hard to follow. I would suggest breaking it down into small sub-sections.

Our response:

Thank you for the valuable suggestion. We agree and have now included following sub-sections into the results section:

3.1. Description of participants

3.2. Parental self-efficacy

3.3. Association of diet quality with family level PSE

3.4. Association of diet quality with mother’s PSE

3.5. Association of diet quality with father’s PSE

Comment: Great work; and very informative tables. I like the way you have presented the findings. Easy to read and indicate the significant results. My only concerns are about your table heading. Table 1; is great with a short title. Table 2- 5 has a long table title. It feels like we are reading a footnote instead of a table. Please use a short table title. Additional information such as ‘adjust for’..’ can be put as the footnote. This will make your hard work look better. Figure 1: Please use a similar approach for Figure 1 too.

Our response:

We appreciate your kind words to us. We have now adjusted the table/figure titles as suggested and included additional information into footnotes.

Comment: The discussion is dense and useful for the readers in the field. Again, consider restructuring the discussion once you re-align results. I think you have done great work on this paper. Consider adding a few lines on how your findings can be used by those who are at the implementation and policy levels.

Our response:

Thank you for these encouraging comments. We have restructured the discussion to follow the same order as the results section. We first start with introducing the most important findings, continue with PSE change between 1.5 and 5 years and then continue similarly as in results with family level PSE, mother’s PSE and father’s PSE level associations.

We have also added a statement about implication for further research into conclusions section.